# Raising Awareness of the Severity of “Contactless Stings” by *Cassiopea* Jellyfish and Kin

**DOI:** 10.3390/ani11123357

**Published:** 2021-11-24

**Authors:** Kaden McKenzie Muffett, Anna M. L. Klompen, Allen G. Collins, Cheryl Lewis Ames

**Affiliations:** 1Department of Marine Biology, Texas A & M University—Galveston, Galveston, TX 77550, USA; 2Ecology and Evolutionary Biology, University of Kansas, Lawrence, KS 66045, USA; annaklompen@ku.edu; 3National Systematics Lab of NOAA Fisheries, Smithsonian National Museum of Natural History, Washington, DC 20013, USA; collinsa@si.edu; 4Department of Invertebrate Zoology, Smithsonian National Museum of Natural History, Washington, DC 20013, USA; 5Graduate School of Agricultural Science, Faculty of Agriculture, Tohoku University, Sendai 981-0952, Japan; ames.cheryl.lynn.a1@tohoku.ac.jp

**Keywords:** envenomation, Scyphozoa, medusa, aquarium, Rhizostomeae, *Cassiopea*, pain

## Abstract

**Simple Summary:**

Current doctrine on jellyfish stings largely focuses on physical contact with a jellyfish. In rhizostome medusae capable of extruding agglomerations of nematocysts within mucus, physical contact is not necessary for skin irritation and pain. Here we highlight pain and symptoms reported by researchers and aquarists working with water around *Cassiopea* and several other jellyfish. We conclude that *Cassiopea*, long thought to be harmless, can lead to multi-day pain and rashes experienced largely as burning and itching sensations along entire limbs. We suggest that recommendations on sting avoidance expand to include consideration of these contactless stings so as to limit a previously under-publicized vector of envenomation.

**Abstract:**

Discussion around avoidance and mitigation of jellyfish stings has traditionally focused on swimmers and divers being mindful of their behavior relative to swimming medusae (pelagic jellyfish). This framework must be restructured with the inclusion of the oblique risk posed by novel autonomous stinging structures like cassiosomes from *Cassiopea* (a jellyfish genus of the taxonomic order Rhizostomeae). Cassiosomes are released by *Cassiopea* sp. into subtropical waters that can consequently sting human skin, causing varying degrees of pain and irritation; this trait extends to other rhizostome jellyfish species. Swimmers and waders may put themselves at risk simply by coming into contact with agitated water in the vicinity of *Cassiopea* medusae, even without touching any part of the jellyfish (medusa, tentacles, or otherwise). Herein, we highlight details provided by 46 researchers and professional aquarists reporting incidents in which they experienced “stinging water” sensations, which we also refer to as “contactless stings’’. We report these findings in order to increase the awareness of a public safety hazard the community may be unaware of in their own labs, aquariums, and sampling locations.

## 1. Introduction

Rhizostome jellyfish species (Cnidaria: Scyphozoa: Rhizostomeae) are united in a well-supported clade [1]. The medusa form of the different rhizostome species possesses multiple mouths as pores distributed along their eight fused oral arms [2], and can be observed in abundance along the coasts of every continent [3,4,5,6,7]. While they have long been understudied from an ecological perspective, they include high impact [8] commercially harvested species [9,10,11], and are common in many areas with substantial ecotourism [12]. While public health authorities advise swimmers to stay away from jellyfish of all kinds, this is often under the supposition that harm from envenomation will arise from physically touching medusae, or more specifically, touching the tentacles or tentacle-like appendages that contain a high density of stinging cell capsules, called nematocysts. However, for some rhizostome jellyfish species, this assumption may be misleading, giving rise to a false sense of security that results in unnecessary irritation or injury due to a phenomenon known as “stinging water”. “Stinging water” occurs when mucus released from the animal into the surrounding water is capable of causing discomfort and pain comparable to contact-based rhizostome jellyfish stings [13]. One family of rhizostome medusae, the Cassiopeidae, or upside-down jellyfish, has been shown to release conglomerations of nematocyst-bearing organized structures suspended in mucus into the surrounding water, causing stinging water sensation or “contactless sting” [13]. Homologous nematocyst-bearing structures, formally called cassiosomes, have been recorded in several additional rhizostomes species, though not examined in great detail [13,14].

Envenomation via nematocyst discharge from unseen jellyfish is not uncommon. Historically, this phenomenon has been assumed a result of stings caused either by tentacle fragments [15], the coronate scyphozoan *Linuche unguiculata* [16], or other small non-cnidarian organisms. Since the 1980s, the rhizostome species *Rhopilema nomadica* has been notorious for causing stings along the Levantine coasts from Egypt to Lebanon. *R. nomadica*’s mild to severe stings result in systematic symptoms (reviewed in Avian et al., 1995 [17]; Glatstein et al. 2018 [18]) that vary in intensity based on human individual sensitivity. Seasonal aggregations of *R. nomadica*, reported to cover expanses of 100 km, are documented in causing mild to severe envenomation [18]. Despite the erection of stinger nets by beach staff to reduce contact with bathers, beachgoers reported a “stinging water” sensation, the cause of which was posited by Glatstein et al., (2018) to be indirect envenomation resulting from contact with portions of severed tentacles and/or mouth arms passing through the net mesh. However, current work suggests that cassiosome-like structures in combination with free nematocysts released together in rhizostome mucus are a likely mechanism triggering or intensifying this type of envenomation event [13]. Throughout this report, we will refer to envenomation symptoms reported in the absence of direct physical contact of human skin with either the tentacles or other attached structures of a jellyfish as “contactless stings”. While not involving medusa contact, these stings do involve contact with nematocyst-bearing structures, and potentially loose, isolated nematocysts released by medusae, and as such could also be referred to as “indirect stings”, “autonomous stings” or “allochthonous stings”.

In conducting this survey and analyzing the results, we endeavored to determine: 1. If broad surface area contactless stings were a common feature of working with *Cassiopea* in an industrial, academic, or recreational context; 2. Whether people that experienced contactless stings (e.g., researchers, aquarium workers, etc.) had experienced this phenomenon with other rhizostomes; and 3. What symptoms were common in these cases.

## 2. Materials and Methods

From December of 2020 to July of 2021, a digital survey aimed at jellyfish researchers and aquarists (corresponding Google Form and Listserv names with email text provided as Appendix A), was distributed through industry-specific electronic mailing lists focused on cnidarian study and husbandry, as well as in a coordinated social media campaign by the authors in December 2020 on Twitter (sample tweet text in Appendix A). The survey, which was made available in six languages (Bahasa, Chinese, English, French, Japanese, and Spanish), was granted an exemption by the Texas A&M University to include questions on “where, when and how” voluntary respondents (researchers and aquarists) had experienced “contactless stings”. Respondents having experienced stinging water multiple times were permitted to report the details of these experiences for up to a total of three instances, indicating the jellyfish taxon responsible in each case (Figure 1). Data from reports that omitted genus-level identification of the jellyfish in question, or responses lacking details surrounding the encounter, were not included in the results reported herein.

Means and standard deviations were computed in R (version 4.0.3) running on RStudio (version 2021.09.0) for pain data. Spearman’s rho was computed using the cor.test function for all reported scaled features related to *Cassiopea* encounters (size, distance, time spent, number of medusae in area) compared against pain scores. All Spearman’s rho results are reported in the Appendix A (See Appendix A).

## 3. Results

### 3.1. General Information

“How long (total) have you worked with *Cassiopea* or other Rhizostome jellyfishes?”“How many times have you felt “stinging water”?“In what role did you experience stinging water?”

Of the 46 survey respondents, 43 (93.5%) had experienced presumed contactless jellyfish stings from *Cassiopea* or another rhizostome jellyfish. The majority of these 43 respondents experienced stinging water sensation while working in their capacity as professional aquarists (58.1%), while the second most common professional cohort (32.6%) was academic researchers. Three people (two aquarists and one researcher) had not experienced this phenomenon at all, though one had worked with *Cassiopea* for less than one year. Others reported having experienced contactless stings as students (undergraduate or graduate) or while engaging in marine recreation activities (Figure 2B). Overall, respondents were primarily long-time handlers of *Cassiopea* medusae. A plurality of respondents worked with *Cassiopea* and other rhizostome species for more than 6 years, while a quarter of respondents had worked with *Cassiopea* for less than a year, or had not researched or handled these medusae professionally (Figure 2A). A supermajority of the respondents had experienced contactless stings on more than three occasions (Figure 2C). In the cases of both research and recreation, the main water-based activity in which contactless stings occurred was snorkeling.

### 3.2. Location

“Where geographically (as specific as possible) did you feel “stinging water?”“What would you classify this location as?”“If you were close to a *Cassiopea*, how close to *Cassiopea* medusae were you?”

Respondents reported experiencing contactless stings along the following coasts: the Caribbean, Florida, Hawaii, French Polynesia, Brazil, Panama, Abu Dhabi, and Israel, as well as within a great number of different public aquaria (Figure 2D). Excluding aquarium tanks, the primary marine ecosystems in which respondents encountered contactless stings were mangroves and lagoons, common habitats for *Cassiopea* upside-down jellyfish species [19]. While those primarily working in public aquariums reportedly experiencing stinging water sensation at a distance of <50 cm from *Cassiopea* medusae, 68% of researchers in the field who encountered this sensation were at a distance of greater than 50 cm, and 47.4% reported being greater than 1 m away from the nearest visible medusa. For in situ accounts, it cannot be guaranteed that researchers did not touch medusae. *Cassiopea,* however, rarely leave the sediment so direct contact stings are less likely overall and especially in those with many years of identification experience. While it is probable that a few of these stings involved some physical contact with medusae, it is likely that given the group surveyed and the robustness and visibility of *Cassiopea* medusae, many did not.

### 3.3. Stinging Water Effects

“What level of discomfort did you experience?”“What would you consider this discomfort most comparable to?”

Survey participants rated their pain on a linear scale of 1 to 5, with 1 as mild irritation, 3 as burning, and 5 as severe pain. The average pain rating for those within five meters of *Cassiopea* medusae was 2.73 (standard deviation of 0.94, range of 1 to 5) (Figure 3). Certain respondents described this pain as most similar to a variety of severe insect bites (e.g., sand flies, mosquitoes or bees), fiberglass burn (described as such by nine respondents), road rash, hives, generalized itching and burning, static shock, or poison ivy (Figure 3).

Redness, itching, and discoloration was reported to last for as little as one hour for some people, and as long as several weeks to months in two cases. Common symptoms of contactless stings noted by three aquarists were bumps (welts) with accompanying redness (contact dermatitis). Graphic photographs provided by one aquarist respondent serve as evidence of the severe effects of stinging water (Figure 4). Respondents who reported specific locations on their bodies where pain and rashes occurred largely noted symptoms on the arms and legs, consistent with contact while reaching into tanks and wading (Figure 5).

Two experienced aquarists working together reported becoming increasingly sensitized to aquarium water housing *Cassiopea* medusae over time; this started with a minimal reaction and escalated to more severe reactions resulting in blistering and pain. No specific question was asked in regard to this phenomenon, so quantitative data on increased sensitivity was not collected, however this phenomenon is in line with known responses to scyphozoan toxins [20]. Hypersensitivity manifesting in skin lesions is reported elsewhere in accounts of envenomation by rhizostome species [15].

The severity of the pain reported by respondents in the wild was not strongly correlated with any one feature queried in the survey to a significant degree (*p* > 0.05). Medusa density and pain (r_s_ = −0.219, *p*-value = 0.367), medusa average size and pain (r_s_ = −0.174, *p*-value = 0.476), highest medusa density and pain (r_s_ = −0.181, *p*-value = 0.457) and distance from highest bloom density and pain (r_s_ = −0.249, *p*-value = 0.372) all had small, statistically insignificant correlations. Respondent time in water around *Cassiopea* had a moderate negative correlation with pain, but was statistically insignificant (r_s_ = −0.492, *p*-value = 0.062). 

The severity of pain reported by respondents in aquariums was also not correlated with recorded circumstances, though the majority of these conditions (i.e., distance from nearest medusa, distance from highest medusa density, highest density of medusae) are not comparable to wild conditions. One aquarium survey respondent noted that pain was only felt when interacting with tanks that included medusae of over 10 cm in diameter. Although our data suggest more intense pain occurs when in the vicinity of larger medusae (Figure 6), this hypothesis was supported, but not with statistical significance, in aquariums (r_s_ = 0.368, *p*-value = 0.121).

### 3.4. Jellyfish Species

Eleven researchers and aquarists reported stinging water incidents from non-*Cassiopea* medusae, this included both rhizostomes and semaeostomes (jellyfishes of another scyphozoan order). The rhizostomes included *Stomolophus* spp. (*n* = 2), *Mastigias papua* (*n* = 1), *Phyllorhiza punctata* (*n* = 1), and *Catostylus mosaicus* (*n* = 4); the latter three were previously reported to produce cassiosomes or cassiosome-like structures [13] (Figure 1). Encounters with these genera were generally rated as less painful than *Cassiopea*, though aquarium cultures reported of these species were in substantially lower densities than the maximum *Cassiopea* densities reported. Notably, stinging water experiences related to *Catostylus* were reported to cause no visible bumps in aquarists who had also experienced stings from *Cassiopea*. These incidents were all in association with physically agitating the water and, therefore, the medusae in their tanks, leading to increased mucus production by the medusae. *Stomolophus* mucus contact in an aquarium setting was reported as less painful than *Cassiopea*, but with the former causing the same skin bump patterning. Disturbance of *Stomolophus meleagris* in situ and the associated mucus released bearing suspended nematocysts has been carefully documented by Shanks and Graham [14]. Similarly, four respondents in this study reported a rash and/or tingling sensation associated with other jellyfish species, either medusae or polyps (the asexual, sessile life stage of many jellyfish species), in laboratory and aquarium settings. No mention of mucus production increase provided in any of these cases (*Aurelia* sp. (2 reports), *Chrysaora* sp. (1), *Chrysaora fuscescens* (1), and *Phacellophora* sp. (1)), suggesting these contactless stings may be triggered by a mechanism different from cassiosomes.

## 4. Discussion

This inaugural survey, intended to determine the extent of contactless stings on researchers and aquarists, was sufficient to provide a snapshot of the severity of the associated stinging water sensation, and suggests that greater caution around these medusae species, both in the field and in culture, is warranted. On the Florida webpage for common jellies found in the NOAA Keys National Marine Sanctuary [21], a major research field station and tourism destination year-round, no mention is made of contactless stings or stinging water due to mucus released by *Cassiopea*, and stings by these jellyfish are listed as generally mild creating a false sense of security for bathers there. In light of our findings, we suggest the implementation of a basic and inexpensive harm-reduction strategy at beaches and marine sanctuaries inhabited by rhizostome jellyfish species, comprising social media messages and physical placards to alert marine recreationalists of the risks of contactless stings. Our findings suggest that the current guidance on in-water sting avoidance (i.e., full coverage wetsuits) [22] is likely insufficient to avoid injury. Several survey respondents explicitly mentioned full-body rashes or full-leg rashes that extended into areas covered by garments or a wet-suit during snorkeling, and one aquarist described developing skin rash from small amounts of water splashing into donned rubber gloves. Many of the stings reported involved minor itching and redness without visible blotchiness; the associated tingling sensation of mild hands within an aquarium tank resembles that of an electric current. The first author of this work has handled *Cassiopea* tanks for over five years non-continuously, and for the first two years failed to recognize the cause of constant arm itchiness and redness brought on by contact with tank water, misattributing the sensation to leaky current (e.g., from a tank heater). Unfortunately, in the absence of an alternative logical explanation, these sorts of incidents have long gone unrecognized as contactless stings due to rhizostome jellyfish mucus. Furthermore, given their relatively low-maintenance husbandry requirements, *Cassiopea* medusae are common in public aquaria, with *C. xamachana* currently a rising jellyfish model in academic institutions [23]. Thus, it is vital that potential volunteers, students, and professionals be properly informed of the potential associated risks of handling water containing rhizostome jellyfish species, trained to mitigate stinging water sensation, and provided with proper medical attention in the event of contactless stings.

Encounters with mild and moderate jellyfish stings are not risk-free [24]. Findings of a long-term study focusing on establishing patterns of toxicity and treatment in children (ages 1–16 years old) stung by *Rhopilema nomadica* conducted during high tourism season (June–August, 2010–2015) revealed that 100% of children experienced pain, erythematous (skin rash due to dilated blood vessels), whip-like linear rash, and burns (mainly first and second degree) and even anaphylaxis (reviewed in Glatstein et al., 2018). Bordehore et al., 2016 found jellyfish stings from Mediterranean jellyfishes, largely mild to moderate stingers, to be the most common injury reported by life guards on beaches along the Spanish Mediterranean coast, accounting for more than half of all injuries in every year from 2008 to 2012 [18]. These injuries signal significant health risks associated with common rhizostome and mild sting species envenomation [24].

Our combined findings, presented here from the literature and survey results represent a strong argument for establishing jellyfish stings, including what we refer to as “contactless stings” as a public health hazard, to be considered under the purview of the World Health Organization’s Control of Neglected tropical diseases (NTD) [25]. If jellyfish stings were lobbied to be included as NTDs, a status venomous snake bites have already acquired, then “policies and strategies” could be established “to enhance global access to interventions for the prevention, control, elimination, and eradication” of this dangerous, and potentially life-threatening, component of the co-habitation of humans and venomous jellyfish.

## 5. Conclusions

Overall, these survey responses suggest that stinging water from *Cassiopea* is commonly experienced by aquarists, researchers, and recreational snorkelers, the symptoms are similar to those caused by mild to moderate jellyfish stings; other species, mainly rhizostome jellyfishes, cause some degree of harm through contactless stings. Therefore, we expect that these findings will contribute to an increased awareness of the problem of contactless stings, and provide guidance on avoiding encounters with stinging water in coastal areas during seasonal occurrences of *Cassiopea* and other rhizostome medusae. Furthermore, with the increase in awareness among the public aquarium community, we trust that guidelines can be modified to include full limb coverage for aquarists handling seawater containing rhizostome medusae in order to mitigate workplace related injury due to contactless stings.

## Figures and Tables

**Figure 1 animals-11-03357-f001:**
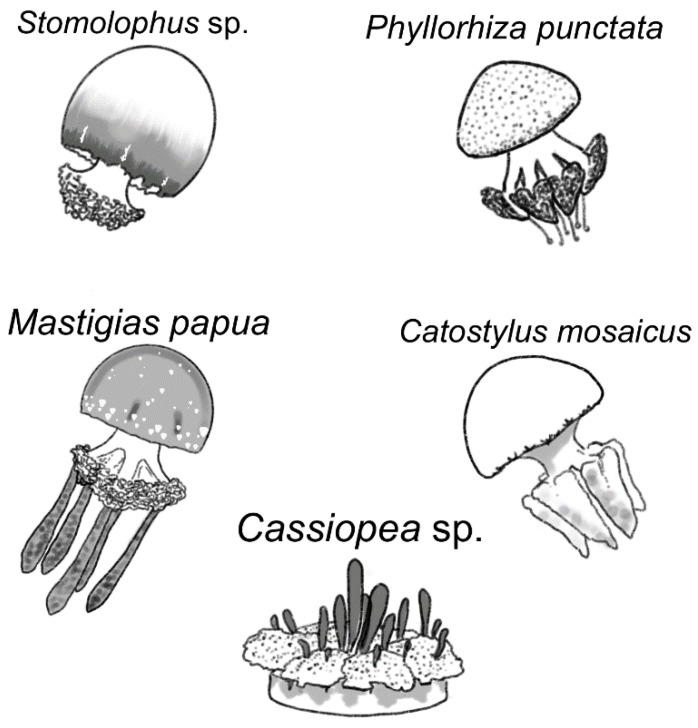
All rhizostome jellyfishes reported in the vicinity of survey respondents reporting “contactless stings” in this study.

**Figure 2 animals-11-03357-f002:**
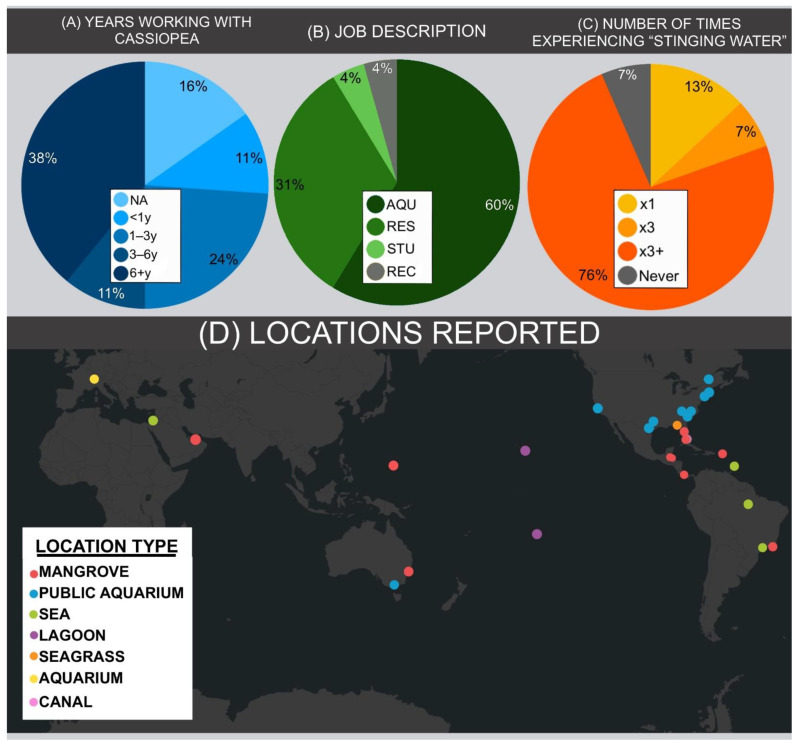
Breakdown of “contactless stings’’ survey responses. (**A**) Number of years respondents have worked with *Cassiopea*, and other rhizostome jellyfishes. Categories are NA (not applicable, has not worked with *Cassiopea*), <1y (less than one year), 1–3y (one to three years), 3–6y (three to six years), or 6+y (greater than six years).(**B**) Capacity in which respondents experienced the resulting stinging water sensation. Categories are AQU (aquarist), RES (researcher), STU (as a non-researcher student) and REC (while engaging in recreational activity). (**C**) Number of times respondents experienced stinging water. (**D**) Geographic distribution of stinging water incidents reported in this study; colored by ecosystem type.

**Figure 3 animals-11-03357-f003:**
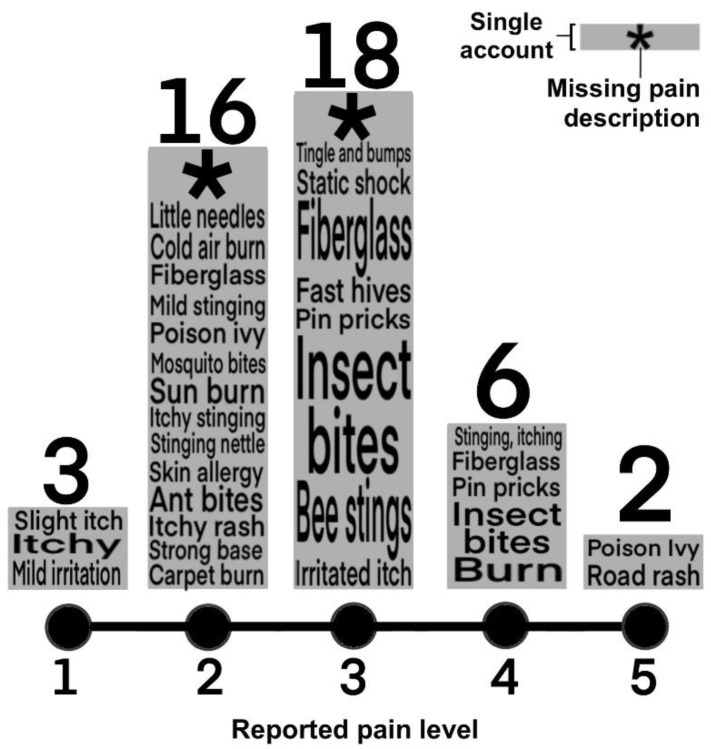
Level of discomfort and descriptors reported for *Cassiopea*-induced stinging water sensation. Participants rated experiences on a pain scale of 1 to 5; numbers appearing above the columns indicate the frequency of responses for each pain rating. Height of descriptor (i.e., “Bee stings”, “Poison ivy”, “Fiberglass”) is based on frequency reported. Descriptors of pain type for each rating were provided by all but four respondents (recorded by asterisks).

**Figure 4 animals-11-03357-f004:**
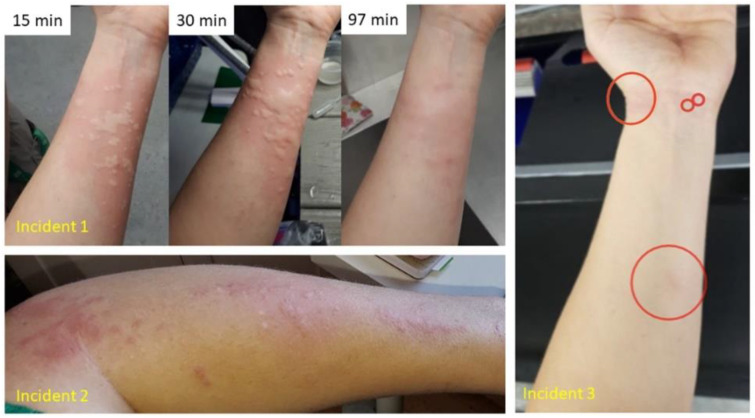
Photographs of an aquarist’s forearm working in a variety of tanks documenting three separate incidents of “contactless sting” injury. (Incident 1): Arms submerged for 15 min in a 10 cm deep sea table tank with *Cassiopea* medusae occurring densely on the substrate. Photographs show effects (welts and dermatitis) of stinging water 15 min, 30 min, and 97 min following removal of the aquarist’s arm from the tank. (Incident 2): Eleven months after incident 1. The same aquarist moved medusae about in a 50 cm deep tank during a 15-min cleaning; the photograph, showing resulting contact dermatitis, was taken several hours later; itching persisted for several days. (Incident 3): Eighteen months after incident 2, the aquarist cleaned the tank for ten minutes, without moving about (agitating) the medusae. Stinging water effects were limited to several small, itchy red marks. Circles indicate irritated areas.

**Figure 5 animals-11-03357-f005:**
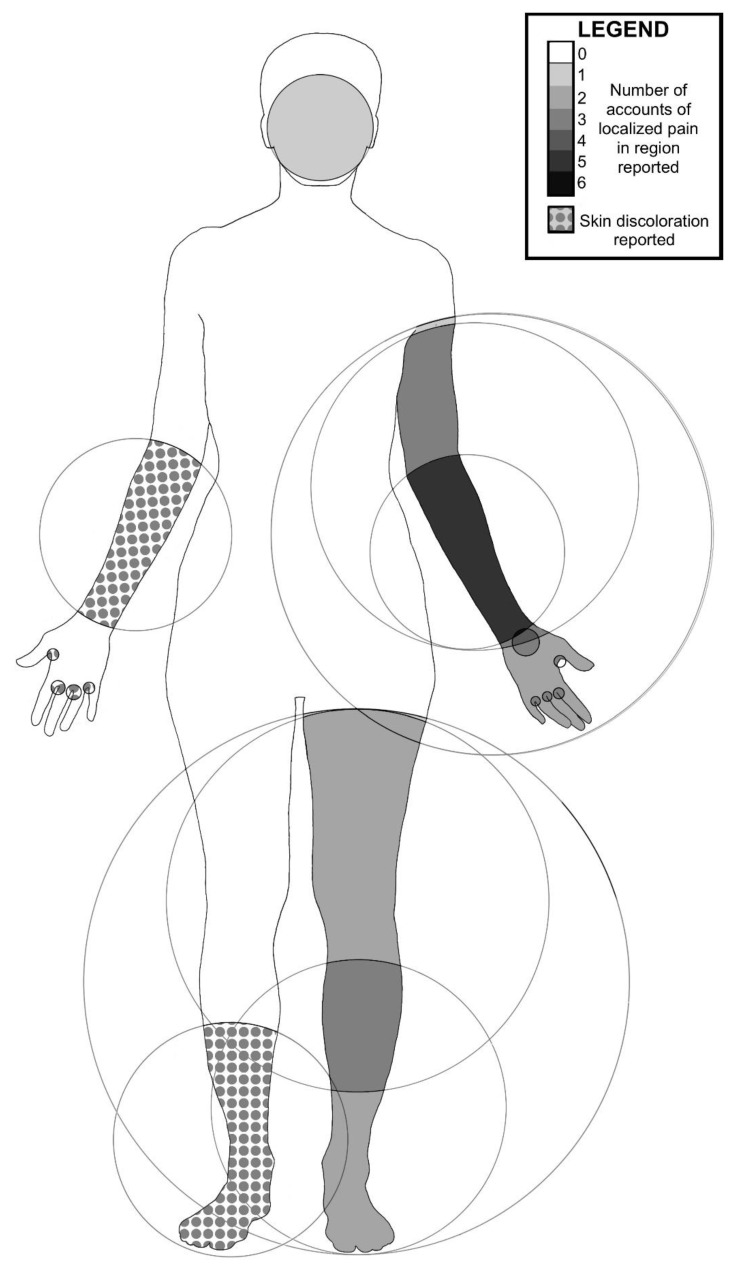
Anatomy of stinging water related pain and rash reported by survey respondents. Darker color indicates a higher number of reports related to pain in specific locations (forearms, feet, and shins). Location on limbs in which people experienced welts and discoloration are patterned with grey circles. Five accounts included pain in the forearms. Upper arms, thighs, and between the fingers were areas of pain in three, two, and three people respectively. Side of body was not reported and is not meaningful in this figure.

**Figure 6 animals-11-03357-f006:**
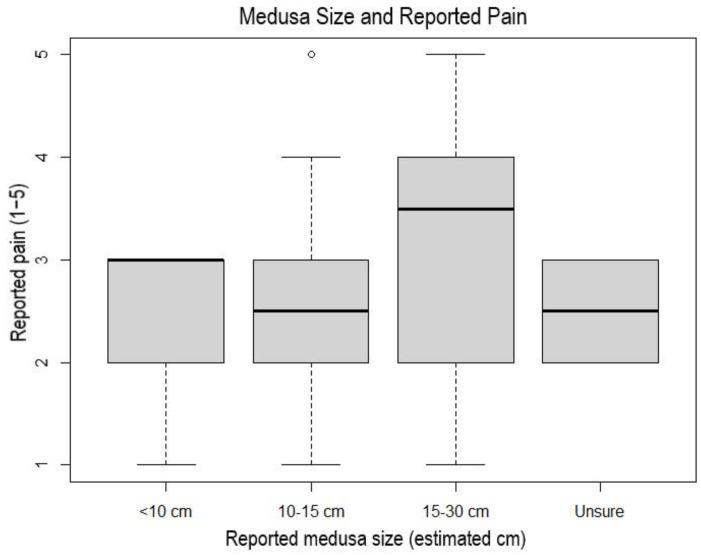
Box-and-whisker plot of stinging water related pain scores (scale of 1–5) in conjunction with size of medusa reported in the vicinity by survey respondents. Pain scores above 3 (burn and irritation) were only reported in instances where medusae in the 10–15 cm size bracket or larger were witnessed in the marine environment. Black central bar represents the mean pain score, gray box represents the middle 50% of pain scores. Dotted lines represent the middle 75% of pain scores at each medusa size. Unfilled dot represents outlier pain scores.

## Data Availability

Anonymized survey answers in spreadsheet form, the survey itself, and the call for participants are all available in the Appendix A.

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
