# Peer review of "Raising Awareness of the Severity of “Contactless Stings” by Cassiopea Jellyfish and Kin"

_animals, 2021, doi:10.3390/ani11123357_

Round 1

Reviewer 1 Report

It is not indicated by the authors of this manuscript whether the reactions to the jellyfish toxins have become more and more intense, as time goes by (months, years), and more stings accumulate.

It is common that the exposure to jellyfish toxins (stings) increases the possibility of allergic reactions. See for example  JA Cañas et al 2018 Jellyfish collagen: a new allergen in the beach Annals of Allergy, Asthma & Immunology    https://doi.org/10.1016/j.anai.2018.01.018  

Could you extract from the responses of the participants this information?

I agree with the authors that the public should minimise the contact  with any  jellyfish species, including those species labelled as "mild sting". A paper that deals with this but regarding to beaches  is C Bordehore et al 2016 Lifeguard assistance at Spanish Mediterranean beaches: Jellyfishprevail and proposals for improving risk management. Ocean & Coastal Management  https://doi.org/10.1016/j.ocecoaman.2016.08.008 

Fig 2.  Regarding stings in the wild not associated with a physical contact with jellyfish, how do you know that it was a contactless sting? For example, it is common with cryptic species (e.g. Cubozoa) that the individual is not seen by the swimmer.

 Minor comments:

Line 25. Please remove a fdot. There are 2 dots in a row after sp..

Line 117. Figure 2 captions should be places below the figure. The letters in black inside the dark blue parts of the graph cannot be seen (low contrast). 

Line 125 please italicise Cassiopea as is is a genus

Author Response

Reviewer #1

  1. It is not indicated by the authors of this manuscript whether the reactions to the jellyfish toxins have become more and more intense, as time goes by (months, years), and more stings accumulate. It is common that the exposure to jellyfish toxins (stings) increases the possibility of allergic reactions. See for example  [JA Cañas et al 2018 Jellyfish collagen: a new allergen in the beach Annals of Allergy, Asthma & Immunology    https://doi.org/10.1016/j.anai.2018.01.018] Could you extract from the responses of the participants this information?

In the previous version of this manuscript, there were only two sentences within the pain section addressing this. This has been separated and expanded with the reviewer’s very useful recommended reading as a citation. Unfortunately, we did not ask any questions to this effect and have only the accounts of two aquarists on this issue. Please refer to lines [158-165] for the relevant content.

  1. I agree with the authors that the public should minimise the contact  with any  jellyfish species, including those species labelled as "mild sting". A paper that deals with this but regarding to beaches  is C Bordehore et al 2016 Lifeguard assistance at Spanish Mediterranean beaches: Jellyfish prevail and proposals for improving risk management. Ocean & Coastal Management  https://doi.org/10.1016/j.ocecoaman.2016.08.008 

The authors agree that this article is relevant and thank the reviewer for this addition. This has been incorporated into the discussion of mild stingers as additional information [252-256].

  1. Fig 2.  Regarding stings in the wild not associated with a physical contact with jellyfish, how do you know that it was a contactless sting? For example, it is common with cryptic species (e.g. Cubozoa) that the individual is not seen by the swimmer.

A note on this issue has been added in the location section [138-143]. While it can’t be determined with certainty whether the respondents’ recolections are correct, we are helped by the biology of Cassiopea, which have thick and very opaque bells and arms and no tentacles. They also sit still on the seafloor unless forcefully disturbed. This combination of factors make unseen medusa contact far less likely than for tentacled medusae of other groups.

  1. Line 25. Please remove a fdot. There are 2 dots in a row after sp..

The authors agree with this revision and have corrected the error.

  1. Line 117. Figure 2 captions should be places below the figure. The letters in black inside the dark blue parts of the graph cannot be seen (low contrast). 

The authors agree with this revision and have corrected the error.

  1. Line 125 please italicise Cassiopea as it is a genus

The authors agree with this revision and have corrected the error.

Reviewer 2 Report

Muffett et al. present a first ever investigation of “contactless stings” by Cassiopea jellyfish and similar organisms that are experienced by biologists in the wild as well as personnel involved in the care of such creatures in aquariums.

The purpose of the work was to increase awareness of this problem, and frankly the reviewer was completely unaware of it prior to reading this paper. Envenomation remote from the organism may be critical to people, especially those with hypersensitivity or a history of anaphylaxis.

The Introduction was informative, and supporting illustrations good – but a little color would not hurt.

The methods were straightforward, but I am not sure I could repeat the procedure. Specifically, the authors stated: “From December of 2020 to July of 2021, a digital survey aimed at jellyfish researchers and aquarists (corresponding Google Form provided as Supplemental Material), was distributed through industry-specific electronic mailing lists focused on cnidarian study and husbandry.” I see the result in one Excel file, but I still do not know how the authors engendered interest from the respondent. Please give more detail concerning these mailing lists.

In figure 6 the authors talk about statistical significance, but in methods no statistical analyses are presented. The data presented seemed more descriptive with a modicum of numeric analysis (mean, SD, range, etc.).  The figure legend for figure 6 does not identify the features of the box-whisker plot. I strongly suggest the authors provide some sort of commentary concerning statistics and data expression in their methods section.

Aside from these easily addressed issues, I enjoyed the article and congratulate the authors on their interesting work.

Author Response

Reviewer #2

  1. The Introduction was informative, and supporting illustrations good – but a little color would not hurt.

While the authors agree that color would likely be more pleasing, the exceptional diversity of color morphs present in even single locations for these medusae mean that choosing one color morph may dramatically alter the appears relative to what someone elsewhere encounters. As this work will hopefully be relevant to people along many coasts and working with many color morphs, we have chosen not to specify colors so as to maintain recognizability across the most situations.

  1. The methods were straightforward, but I am not sure I could repeat the procedure. Specifically, the authors stated: “From December of 2020 to July of 2021, a digital survey aimed at jellyfish researchers and aquarists (corresponding Google Form provided as Supplemental Material), was distributed through industry-specific electronic mailing lists focused on cnidarian study and husbandry.” I see the result in one Excel file, but I still do not know how the authors engendered interest from the respondent. Please give more detail concerning these mailing lists.

The authors agree that too little information was provided on this. A PDF of the text provided in the survey announcement emails is now included in the Supplementary Materials, as well as a list of list servs through which the survey was distributed. This has been added to the methods. The authors thank the reviewer for bringing to light this oversight.

  1. In figure 6 the authors talk about statistical significance, but in methods no statistical analyses are presented. The data presented seemed more descriptive with a modicum of numeric analysis (mean, SD, range, etc.).  The figure legend for figure 6 does not identify the features of the box-whisker plot. I strongly suggest the authors provide some sort of commentary concerning statistics and data expression in their methods section.

The authors agree with this criticism and a statement on the statistics done has been added to the methods [94-97] and the Figure 6 caption has been updated for clarity as well.

  1. Aside from these easily addressed issues, I enjoyed the article and congratulate the authors on their interesting work.

We thank the reviewer for their kind words of support.

Reviewer 3 Report

This study is exciting and completely changes the approach of a clinical toxicologist can have on jellyfish stings. The text should be published. Some propositions to improve the text:

  • We all understand that it is very complicated to give a name to this new kind of envenomation. However I am not sure that "contactless stings" is appropriate because there is no contact with the jellyfish but there is a contact with toxic cells or toxic parts of the jellyfish. This must be discussed in the text.
  • The Figures 3 and 5 are not really well presented. Is it possible to improve the quality of the presentation?

Author Response

Reviewer #3

  1. This study is exciting and completely changes the approach of a clinical toxicologist can have on jellyfish stings. The text should be published.

We thank the reviewer for their kind words of support.

  1. We all understand that it is very complicated to give a name to this new kind of envenomation. However I am not sure that "contactless stings" is appropriate because there is no contact with the jellyfish but there is a contact with toxic cells or toxic parts of the jellyfish. This must be discussed in the text.

The authors agree with this criticism and have clarified the introduction to make clear the nature of the contact and offer alternative possible names for this phenomenon [75-79]. As there is currently not experimental validation on the nature of the specific stings discussed, we will retain the most general term “contactless stings”, with the hope that a less simplistic terms are used once full validations of the phenomenon discussed are released.   

  1. The Figures 3 and 5 are not really well presented. Is it possible to improve the quality of the presentation?

The authors agree that these figures could be improved. Figures 3 & 5 have been replaced with clearer, cleaner versions.

Round 2

Reviewer 2 Report

I appreciate the effort made by the authors. However, they still do not present the software used for statistical analysis. Rho is presented as an r. The statistical significance of a relationship is presented as P, and a P<0.05 is considered significant. The authors need to seek statistical assistance if this difficult for them to complete.

Author Response

I appreciate the effort made by the authors. However, they still do not present the software used for statistical analysis. Rho is presented as an r. The statistical significance of a relationship is presented as P, and a P<0.05 is considered significant. The authors need to seek statistical assistance if this difficult for them to complete.

I am sorry that the explanation for statistical methods was unclear. The Spearman’s rho correlation coefficient was calculated in the software R (version 4.0.3) using the cor.test code included below for you.

The confidence level (a=0.05) is my preference, however if you feel strongly that given the low number of includible accounts (only those not at an aquariums) a 0.1 level would be more suitable we are willing to make this change. More stringent alpha values (a=0.01) seem unsuitable for the data amount and type. The version name for R and RStudio are now in the text as well as “cor.test”.

Reference to this alpha threshold has been removed from the methods and placed on line 198, at the point in which the lack of significance is mentioned, where it is more meaningful.

If the reviewer would prefer Kendall’s tau is used to reduce the impact of ties, we are open to replacing Spearman’s rho, in this data they perform similarly.

If either the threshold was lowered or Kendall’s tau was used, the “length of time in the area” would be mentioned in the text as significant.
